# A Novel AURKA Mutant-Induced Early-Onset Severe Hepatocarcinogenesis Greater than Wild-Type via Activating Different Pathways in Zebrafish

**DOI:** 10.3390/cancers11070927

**Published:** 2019-07-02

**Authors:** Zhong-Liang Su, Chien-Wei Su, Yi-Luen Huang, Wan-Yu Yang, Bonifasius Putera Sampurna, Toru Ouchi, Kuan-Lin Lee, Chen-Sheng Wu, Horng-Dar Wang, Chiou-Hwa Yuh

**Affiliations:** 1Institute of Molecular and Genomic Medicine, National Health Research Institutes, Zhunan, Miaoli 35053, Taiwan; 2Institute of Biotechnology, National Tsing Hua University, Hsinchu 30013, Taiwan; 3Division of Gastroenterology and Hepatology, Department of Medicine, Taipei Veterans General Hospital, Taipei 11217, Taiwan; 4Department of Cancer Genetics, Roswell Park Cancer Institute, Buffalo, NY 14263, USA; 5Department of Biological Science and Technology, National Chiao Tung University, Hsinchu 30010, Taiwan; 6Institute of Bioinformatics and Structural Biology, National Tsing Hua University, Hsinchu 30013, Taiwan; 7Ph.D. Program in Environmental and Occupational Medicine, Kaohsiung Medical University, Kaohsiung 80708, Taiwan

**Keywords:** hepatocellular carcinoma (HCC), zebrafish, Aurora A kinase (AURKA), β-catenin, AKT signaling pathway

## Abstract

Aurora A kinase (AURKA) is an important regulator in mitotic progression and is overexpressed frequently in human cancers, including hepatocellular carcinoma (HCC). Many AURKA mutations were identified in cancer patients. Overexpressing wild-type Aurka developed a low incidence of hepatic tumors after long latency in mice. However, none of the AURKA mutant animal models have ever been described. The mechanism of mutant AURKA-mediated hepatocarcinogenesis is still unclear. A novel AURKA mutation with a.a.352 Valine to Isoleucine (V352I) was identified from clinical specimens. By using liver-specific transgenic fish overexpressing both the mutant and wild-type AURKA, the AURKA(V352I)-induced hepatocarcinogenesis was earlier and much more severe than wild-type AURKA. Although an increase of the expression of lipogenic enzyme and lipogenic factor was observed in both AURKA(V352I) and AURKA(WT) transgenic fish, AURKA(V352I) has a greater probability to promote fibrosis at 3 months compared to AURKA(WT). Furthermore, the expression levels of cell cycle/proliferation markers were higher in the AURKA(V352I) mutant than AURKA(WT) in transgenic fish, implying that the AURKA(V352I) mutant may accelerate HCC progression. Moreover, we found that the AURKA(V352I) mutant activates AKT signaling and increases nuclear β-catenin, but AURKA(WT) only activates membrane form β-catenin, which may account for the differences. In this study, we provide a new insight, that the AURKA(V352I) mutation contributes to early onset hepatocarcinogenesis, possibly through activation of different pathways than AURKA(WT). This transgenic fish may serve as a drug-screening platform for potential precision medicine therapeutics.

## 1. Introduction

Hepatocellular carcinoma (HCC) is the fifth most common cancer and ranks as the third leading cause of mortality worldwide [1,2], and there is still no effective therapy available due to its heterogeneity [3]. In the era of precision medicine, to understand the real causative factors underlying the carcinogenesis, and provide treatment with precise personalized medicine, is extremely important and remains the most effective therapy for the HCC [4]. Currently, a multikinase inhibitor, Sorafenib, is the only U.S. Food and Drug Administration (FDA)-approved drug to treat advanced-stage HCC patients [5]. Other drugs or combinations of chemotherapeutic agents are urgently required. Genome-wide association study (GWAS) technology enables the discovery of many genetic risk factors and mutations related to HCC [6]. However, animal models for functional analysis are warranted to confirm the roles of those mutations in hepatocarcinogenesis and develop precision medicine against those novel mutants.

The zebrafish is a vertebrate model with higher relevance to humans. Approximately 70% of human genes have at least one obvious zebrafish orthologue [7]. Well-developed gene transfer technology has boosted zebrafish as a prevalent research model in different research fields, including human diseases, cancer study, and drug screening [8,9,10]. Zebrafish provide an in vivo model organism for cancer research, drug identification, validation, and screening [11]. Zebrafish cancer models could be part of preclinical precision medicine approaches [12]. Even in the Cancer Moonshot project, the zebrafish plays an important role in “developing new cancer therapeutic technologies” [13]. Previously, our lab has used transgenic zebrafish to study hepatocarcinogenesis and drug screening [14,15,16,17,18]. In this study, we aim to explore the function of wild-type and mutant AURKA in liver cancer formation.

The aurora kinase family is a master regulator of mitotic progression and frequently overexpressed in human cancers. The aurora kinase family consists of Aurora A, Aurora B, and Aurora C [19], which are evolutionally conserved from yeast to human [20,21]. Aurora A kinase (AURKA) plays an important role at the late G2 phase for centrosome maturation and mitotic commitment, especially in the process of centrosome duplication, bipolar spindle formation, chromosomal segregation, and cytokinesis [22,23,24,25]. The timing and amplitude of AURKA must be well controlled for the progression of mitosis. Aurora A kinase is activated by autophosphorylation through interaction with TPX2, the best-known substrate and activator of AURKA [26] during mitosis. In late mitosis-G1, the level of AURKA decreases through ubiquitin-dependent degradation mediated by ubiquitin ligase and the anaphase promoting complex (APC) [19,27]. Aberrant expression of AURKA may lead to genetic instability [28] and cause development of many cancers [27]. The novel non-canonical role of AURKA in DNA replication has been revealed [29], promoting tumor progression, including inducing phosphorylation of Akt and mammalian target of the rapamycin (mTOR), phosphorylation of mitogen-activated protein kinase (MAPK), and activation of epithelial–mesenchymal transition (EMT) reprogramming, making AURKA an attractive target for cancer therapeutics [28].

Sixty percent (60%) of HCC cases showed an increase of AURKA mRNA and protein levels. Aurora A kinase overexpression more frequently occurred in higher HCC grades and stages, indicating that AURKA is associated with HCC development and progression [30]. However, AURKA genomic amplification was detected in only 3% of HCC, indicating that other mechanisms are involved in AURKA activation. Aurora A kinase activates EMT and reprograms cancer cell stemness, which is responsible for metastases in head and neck cancer, breast cancer, and HCC [31,32,33]. Furthermore, overexpression of AURKA in mouse embryonic fibroblasts (MEF) cells did not show the transformed phenotype [34] and in transgenic liver led to a low incidence (3.8%) (2 of 52) of hepatic tumor formation after a long latency period in mouse models [35], clearly indicating that AURKA overexpression alone is insufficient to induce carcinogenesis.

Previously, it was reported that the AURKA Ile31Phe mutation might play a role in mediating susceptibility to HBV-related HCC among Chinese people [36]. In the Han Chinese population, four Single Nucleotide Polymorphisms (SNPs) in AURKA were associated with breast cancer susceptibility [37], and another haplotype of AURKA was also associated with increased risk of endometrial carcinoma [38]; in Taiwan, AURKA SNP is also associated with oral cancer development [39]. These SNPs are not only involved in carcinogenesis but also have a predictive value of the efficiency of cetuximab treatment in head and neck squamous cell carcinoma [40].

From clinical samples, we identified a novel AURKA mutation on a.a.352 from Valine to Isoleucine, in addition to AURKA overexpression. This amino acid is located in the beta strand and is conserved between human and zebrafish, indicating that it has an important function. We hypothesize that the AURKA(V352I) mutation plays a role in liver cancer formation and drug responsiveness.

The phosphatidylinositol 3-kinase (PI3K)/Akt/phosphatase and tensin homologue deleted on the chromosome 10 (PTEN)/mammalian target of the rapamycin (mTOR) pathway is an intracellular signaling pathway important in regulating the cell cycle [41,42,43]. The PI3K/Akt/PTEN/mTOR signaling pathway is directly related to cellular quiescence, proliferation, and cancer. The PI3K is a class of cooperating molecules that are distinguished from their structure, regulation, and function. Class IA PI3Ks are the most studied in cell division and transformation [44]. The AKT activation is phosphorylated by PI3K and further directly phosphorylates mTOR, localizing it in the plasma membrane [42]. The AKT plays a critical role in the regulation of multiple biological processes including cell survival, proliferation, apoptosis, and glycogen synthesis [45]. The mammalian target of rapamycin is a Ser/Thr protein kinase and acts as a sensor for ATP and amino acid, responsible for balancing the availability of nutrients and cell growth. It plays a key role in cellular growth and homeostasis, and its regulation is frequently altered in tumors [46]. The phosphatase and tensin homologue deleted on chromosome 10 is involved in a wide variety of human cancers and is a major negative regulator of the PI3K/Akt signaling pathway [47].

It was revealed that AURKA promoted metastasis through the PI3K/AKT pathway [31]. Aurora A kinase promotes expression of the nuclear Ikappaβ-alpha (Iκβα) protein and enhances nuclear factor kappa light chain enhancer of activated B cells (NF-κB) activity, thus promoting the transcription of miR-21, which negatively regulates PTEN and then inhibits caspase-3-mediated apoptosis. This AURKA/NF-κB/miR-21/PTEN/Akt signaling axis was involved in AURKA promoting chemoresistance in HCC [48]. In addition to the above pathways, AURKA is known to be involved in many other pathways and form signaling networks in cancers, including tumor suppressor p53, tumor protein p73, mouse double minute 2 homolog (MDM2), Glycogen synthase kinase 3 beta (GSK3b), v-myc myelocytomatosis viral related oncogene (N-MYC), proto-oncogene tyrosine-protein kinase (SRC), signal transducer and activator of transcription 3 (STAT3), and breast cancer gene 1/2 (BRCA1/BRCA2) [49].

In this study, we demonstrated that the wild-type AURKA and the V352I mutant can differentially activate those signaling axes in transgenic fish during hepatocarcinogenesis, which provides a new insight to AURKA-mediated liver tumorigenesis.

## 2. Results

### 2.1. Overexpression of AURKA is Dramatically Increased in Transgenic Zebrafish Compared to the Control Zebrafish

The mammalian Aurora kinase family are master regulators of mitotic progression and are frequently overexpressed in human cancers, including HCC. The level of AURKA is well correlated with a high grade and stage of HCC, indicating that overexpression of AURKA plays a role in the development and progression of HCC. From clinical specimens, the AURKA mutation on a.a.352, from Valine to Isoleucine (V352I), was identified (Appendix A). We created transgenic fish lines to test whether V352I mutant AURKA might play a different role than WT AURKA in hepatocarcinogenesis.

The hepatocarcinogenesis of the AURKA(V352I) and AURKA(WT) transgenic fish were examined at 3, 5, 7, 9, and 11 months. At different stages, we collected zebrafish liver from ten individual fish and analyzed the expression of marker genes using qPCR, checked the histopathological features using hematoxylin and eosin stain (HE stain), and examined the protein levels of proliferating cell nuclear antigen (PCNA), β-catenin, PTEN, p-Akt, and p-mTOR using immunohistochemistry staining.

We first examined the mRNA level of AURKA in transgenic fish using primers specific to human AURKA (Figure 1). Compared to the control fish, AURKA was dramatically overexpressed in transgenic fish. Although the AURKA mRNA expression level in AURKA(WT) transgenic fish (Figure 1A) was 3–4 fold lower than mutant AURKA transgenic fish (Figure 1B) at 3 M, the levels are similar at 5 M, 7 M, and 11 M, and interestingly, the level was 7-fold lower in AURKA(V352I) compared to AURKA(WT) at 9 M. Nevertheless, the transgenic fish overexpressed human AURKA compared to the control fish, so we could further compare those two transgenic fish for lipogenesis, fibrosis, and cancer formation.

### 2.2. The Expression of Lipogenic Enzyme and Lipogenic Factor Were Observed in Both AURKA(V352I) and AURKA(WT) Transgenic Fish

Previous studies have shown that hepatocarcinogenesis starts with steatosis (fatty liver) in human and many transgenic fish. Therefore, we examined the expression of lipogenic transcription factors, including peroxisome proliferator-activated receptor gamma (*pparγ*), sterol regulatory element-binding transcription factor 1 (*srebp1*), and the carbohydrate-responsive element-binding protein (*chrebp*). We also examined the expression levels of the lipogenic enzymes, including diacylglycerol *O*-Acyltransferase 2 (*dgat2*), phosphatidate phosphatase (*pap*), and fatty acid synthase (*fasn*) from 3, 5, 7, 9, and 11 months of age. The stage-matched Tg(fabp10a:HBx-mCherry) was used as a control, and expression fold versus control was calculated.

Using qPCR analysis, we found that the lipogenic factors were overexpressed in both AURKA transgenic fish compared to the control fish. The expression of *pparγ* was upregulated at 7 months in AURKA(WT) transgenic fish and was upregulated as early as 3 months in AURKA(V352I) transgenic fish (Figure 2A,D). The expression of *srebp1* (Figure 2B,E) and *chrebp* (Figure 2C,F) increased at 5 months and 3 months for AURKA(WT) and AURKA(V352I) transgenic fish, respectively. Furthermore, the fold of overexpression in AURKA(V352I) was much higher than in AURKA(WT) transgenic fish. To validate the results of the AURKA(V352I) transgenic fish, we used another AURKA(V352I) mutant transgenic line (TG2) for qPCR analysis. The TG2 had higher AURKA expression (Appendix A) and exhibited a much higher *pparγ* expression level compared to TG1 (Appendix A). Although we observed that *pparγ* expression is higher in mutant, other lipogenic factors (*srebp1* and *chrebp*) did not appear to show any notable differences between the AURKA(WT) and AURKA(V352I); this result was also confirmed by TG2.

The expression level of the lipogenic enzymes for the AURKA(WT) and AURKA(V352I) transgenic fish were also compared. The expressions of *dgat2* (Figure 3A,D) and *pap* (Figure 3B,E) were upregulated in both AURKA transgenic fish at 3 months. However, the expression of *fasn* in both AURKA transgenic fish was upregulated at 11 months (Figure 3C,F). There don’t appear to be any notable differences between WT and the V352I mutant in the expression of lipogenic enzymes *pap* and *dgat2*, which was independently confirmed by another V352I mutant (Appendix A).

In the V352I mutant, the expression of lipogenic enzyme and lipogenic factor increased at 3 months. In WT, the expression of lipogenic enzyme and lipogenic factor increased from 5 to roughly 7 months. In summary, our results suggest that overexpression both AURKA(WT) and AURKA(V352I) induced expression of lipogenic factors and enzymes, indicating the AURKA(WT) and AURKA(V352I) can induce steatosis.

### 2.3. AURKA(V352I) Has More Probability in Promoting Fibrosis at 3 Months

Because 90% of HCC cases have a natural history of unresolved inflammation and severe fibrosis (or cirrhosis), we then examined the fibrosis marker genes, including collagen type I alpha 1 (*cola1a1*), connective tissue growth factor (*ctgfa*), and heparanase (*hpse*) in the AURKA transgenic fish. Overexpression of *cola1a1* was found at 5 months in the AURKA(WT) transgenic fish (Figure 4A) but was upregulated at 3 months in AURKA(V352I) transgenic fish (Figure 4D). The expression of *ctgfa* and *hpse* in both transgenic fish was upregulated from 5 to 7 months (Figure 4B,C,E,F). The fold of fibrosis marker genes showed no difference between AURKA(WT) and AURKA(V352I) transgenic fish. Another AURKA(V352I) mutant transgenic line (TG2) also showed the same expression pattern of fibrosis markers (Appendix A). Compared with the expression data of lipogenic factor and enzyme, the fold of the fibrosis marker gene (*cola1a1*) is higher than in lipogenic-related genes. The data suggest that HCC induced by AURKA overexpression might be promoted through fibrosis pathology.

### 2.4. Expression of Cell Cycle-Related Genes/Proliferation Markers Were Significantly much Higher and Earlier in AURKA(V352I) than in AURKA(WT) Transgenic Zebrafish

Unlimited replicative potential is one of the hallmarks of cancer cells. Because cancer cells have the properties of uncontrolled cell proliferation and dysfunction of cell cycle checkpoint, we analyzed the cell cycle-related genes/proliferation marker genes G1/S-specific cyclin-E1 (*ccne1*), cyclin-dependent kinase 1 (*cdk1*), and cyclin-dependent kinase 2 (*cdk2*) by qPCR. In AURKA(WT) transgenic fish, we only observed a slightly increase of *ccne1* at 9 months (Figure 5A), increased expression of *cdk1* at 7 months (Figure 5B), and significant upregulation of *cdk2* at 5 months (Figure 5C). However, in AURKA(V352I) transgenic fish, the expression of *ccne1* was upregulated at 5 and 7 months (Figure 5D), there was a significant increase of *cdk1* expression at 7 months (Figure 5E), and there was a dramatic increase of *cdk2* at 5 months of age (Figure 5F). Surprisingly, the fold of the upregulated cycle-related genes/proliferation marker genes in AURKA(V352I) transgenic fish was greater than in AURKA(WT). This result implies that cell proliferation in AURKA(V352I) transgenic fish might be more severe than in AURKA(WT). Another AURKA(V352I) mutant transgenic line (TG2) also showed the same expression pattern of cell cycle/proliferation markers (Appendix A).

### 2.5. Hematoxylin and Eosin Staining Reveals that AURKA(V352I) Promotes HCC at 7 Months, Whereas AURKA(WT) Promotes HCC Later at 9 Months

After analyzing the expression of lipogenic factor and enzyme, fibrosis marker genes, and cell cycle-related genes via qPCR, we examined the histopathological features of the liver tissue from AURKA(WT) and AURKA(V352I) transgenic fish using an HE stain (Figure 6). The criteria of different pathology patterns were previously published from our lab [50]. Steatosis was observed from 3 months—atypical pathological features such as hyperplasia, dysplasia, and HCC. The AURKA(WT) fish developed HCC at 9 months of age, and about 30% of the fish were diagnosed as HCC positive while the rest had hyperplasia. The AURKA(V352I) fish had dramatic HCC progression at 7 months, and 50% of the fish developed HCC. The tumors in transgenic zebrafish may have regressed due to the self-healing of zebrafish, which was observed in many transgenic fish lines from our lab and others [16,18,50,51]. Compared with the AURKA(V352I) fish, HCC progression was not so significant in AURKA(WT) fish, and the possibility of abnormal cell morphology evenly existed in all stages.

### 2.6. Proliferating Cell Nuclear Antigen Staining Indicated that Cell Proliferation in AURKA(V352I) Was More Severe than AURKA(WT) and Control Fish from 3 to 7 Months

To evaluate cell proliferation, we performed the immunohistochemistry for PCNA. Immunohistochemistry scoring was graded based on two parameters: the intensity grade (score: 1–3) and the proportion of positive tumor cells (score: 1–4) [52]. The immunoreactive score (IRS) was obtained by multiplying the intensity grade by the positive proportion score [53] (Figure 7). We found that the IRS score of PCNA was higher in AURKA fish than in control fish at 5 and 7 months. In particular, the IRS score of PCNA in AURKA(V352I) transgenic fish was significantly higher than in AURKA(WT) at 5 months. This result confirms that AURKA(V352I) transgenic fish has more dramatically severe hepatocarcinogenesis than AURKA(WT) transgenic fish.

### 2.7. Less Membrane Bound β-Catenin for Metastatic Behavior of Hepatocyte in AURKA(V352I) Transgenic Fish for EMT Transition

According to the previous study, mutations and overexpression of β-catenin are associated with many cancers, including hepatocellular carcinoma [54,55]. As mentioned previously, we multiplied the intensity grade by the positive proportion to get the immunoreactive score (IRS). We found that the IRS of immunostaining for β-catenin in AURKA(WT) fish was always higher than in AURKA(V352I) fish at 3, 5, and 9 months and comparable to the control fish at these stages (Figure 8). Compared with the AURKA(WT) and control fish, however, it is unexpected that the AURKA(V352I) fish had a constantly low IRS score of immunostaining for β-catenin (Figure 8B). This might suggest the presence of less membrane bound β-catenin and that the actual transcriptional activity of β-catenin is high in AURKA(V352I) transgenic fish. To further clarify this point, we scored the nuclear β-catenin and found that the nuclear β-catenin level was, indeed, significantly higher at all stages in AURKA(V352I) transgenic fish than in the AURKA(WT) and control fish (Figure 8C).

### 2.8. The pAKT in AURKA(V352I) Was Significantly Higher than AURKA(WT) and Control Fish in all Stages

Based on a previous study, it has been shown that AURKA is a potential oncogene in mammary gland tumors in mice through activation of the Akt-mTOR pathway [56]. In liver cancer cell lines, overexpression of AURKA can reduce PTEN and increase the phosphorylation of Akt and mTOR to induce cell proliferation and transformation. We used immunostaining staining to examine whether the Akt-mTOR pathway was activated in the AURKA transgenic fish HCC model (Figure 9, Figure 10 and Figure 11).

We found that the PTEN decreased in AURKA(V352I) at 7 months of age, but this decrease was not significant (Figure 9). The IRS score of p-AKT in AURKA(V352I) was significantly higher than in AURKA(WT) and the control fish in all stages (Figure 10). There was no obvious difference for the p-mTOR between AURKA(V352I) and AURKA(WT), except that at 11 months it was significantly higher in AURKA(WT) (Figure 11). These results indicate that the p-AKT is the main player for AURKA(V352I), causing more severe hepatocarcinogenesis than in AURKA(WT).

### 2.9. Multiple Kinase Inhibitors Cannot Prevent Cancer Formation in AURKA(WT) and AURKA(V352I) Transgenic Fish

Due to genomic heterogeneity, there is no effective drug to treat HCC. Hepatitis B virus X protein (HBx) induced HCC formation via activating Src expression [57]. Overexpression of *HBx* and *src* in a p53 mutant induced HCC formation in transgenic fish [43]. Sorafenib, a multikinase inhibitor, is the only FDA-approved drug to treat advanced stage HCC patients [5]. Sorafenib and two other multiple kinase inhibitors (BPR1J419S1 and BPR1J420S1) have been proved to prevent HCC in *HBx*, *src*, and *p53^−/+^* transgenic fish [58]. To understand whether HCC developed by AURKA overexpression can be cured by Sorafenib, BPR1J419S1, and BPR1J420S1, we fed the 5-month-old transgenic fish by gavage feeding with these drugs for one month, as published [58]. We then sacrificed the fish and examined the cell cycle/proliferation markers using qPCR. We found that multiple kinase inhibitors cannot prevent cancer formation in AURKA(WT) and AURKA(V352I) transgenic fish (Appendix A). The development of precision medicine targeting the AURKA overexpression and novel mutants is still ongoing.

## 3. Discussion

Zebrafish is a fantastic model system for studying genetic mutations found in hepatocellular carcinoma. Many transgenic zebrafish HCC models have been established to study recurrent and novel driver mutations [59]. Aurora A kinase is cell cycle-regulated serine/threonine kinase, involved in microtubule formation and stabilization at the spindle pole during chromosome segregation. Overexpression of AURKA is found in many cancers. However, AURKA is a low-penetrance tumor-susceptibility gene in mice and humans. In this study, we demonstrated a novel AURKA mutation (that changes a.a.352 from Valine to Isoleucine) induced early-onset severe hepatocarcinogenesis greater than wild-type AURKA by activating the p-AKT pathway and increasing the nuclear β-catenin protein levels.

The lipogenic transcription factor, *pparγ*, which involves fatty acid storage and glucose metabolism [60], was upregulated in both AURKA(WT) and AURKA(V352I) transgenic fish compared to the control fish. The expression of *pparγ* in AURKA(V352I) transgenic fish was upregulated at 3 months. However, the expression of *pparγ* in AURKA(WT) transgenic fish was upregulated at 5 months. Additionally, the expression of *srebp1* and *chrebp*, transcription factors responsible for the synthesis of enzymes involved in sterol biosynthesis [61] and coupling hepatic glucose utilization and lipid synthesis [62,63] were increased at 5 months in both AURKA(WT) and AURKA(V352I) transgenic fish. Furthermore, we got consistent results for the lipogenic enzymes, *dgat2* and *pap*, involved in the regulation of triglyceride synthesis, lipid synthesis, and metabolism [64,65]; they were upregulated at 3 months in both AURKA(WT) and AURKA(V352I) transgenic fish. However, the expression of *fasn*, a key enzyme involved in de novo lipogenesis converting excess carbon intake into fatty acids for storage [66], in both AURKA transgenic fish, was upregulated at a much later stage. From the HE stain, we confirmed that both AURKA(WT) and AURKA(V352I) can induce steatosis in transgenic fish as early as 3 to 5 months.

Overexpression of AURKA(V352I) induced the expression of the fibrosis marker *col1a1* at 3 months, whereas AURKA(WT) upregulated *col1a1* at 5 months from the qPCR analysis of fibrosis markers. The expression of *ctgfa* and *hpse* was upregulated from 5 to 7 months in both AURKA(WT) and AURKA(V352I) transgenic fish. Both AURKA(WT) and AURKA(V352I) could induce fibrosis in transgenic fish from 5 to 7 months, which indicates that AURKA(V352I) is a very potent oncogene in hepatocarcinogenesis. From the AURKA(WT) and AURKA(V352I) transgenic fish, we observed that the expression of lipogenic factors/enzyme and fibrosis markers was downregulated at later time points. These results were observed from many other transgenic fish overexpressing oncogenes in the liver [14,15,16,17,50]. Those expression patterns were due to the progression of hepatocarcinogenesis. Those fish will develop steatosis and fibrosis at earlier time points, accompanied with the upregulation of lipogenic factors/enzyme and fibrosis markers, and will develop into HCC at later time points, accompanied by the upregulation of cell cycle/proliferation markers and the downregulation of lipogenic factors/enzyme and fibrosis markers.

Moreover, from the qPCR data of cell cycle/proliferating markers and PCNA immunostaining, we found that AURKA(V352I) induces more significant cell proliferation than AURKA(WT) and resulted in earlier and higher incidence of HCC formation than AURKA(WT). We are curious about the differences in the oncogenic mechanisms of AURKA(V352I) and AURKA(WT). Aurora A kinase promoted metastasis through the PI3K/AKT pathway [31] and participated in the Akt/mTOR pathway [56]. We found that PTEN and p-mTOR showed no differences between AURKA and the control fish. Despite similar PTEN and p-mTOR expression levels based on their immunostaining score, p-Akt expression level was significantly higher in AURKA(V352I) than in AURKA(WT). Therefore, we assume that Akt is the driver gene of AURKA(V352I) oncogenesis.

It was already known that β-catenin phosphorylation by Akt could promote β-catenin transcriptional activity and lead to tumor invasion and development [67]. We found that total β-catenin expression is lower in AURKA(V352I) than AURKA(WT). Moreover, the nuclear β-catenin is significantly higher in AURKA(V352I) than in AURKA(WT) and the control fish. Interestingly, AURKA(WT) exhibits higher cytoplasmic β-catenin than the control fish. β-catenin is one of the primary oncogenes involved in HCC development [54]. The subcellular localization of β-catenin determines the function of β-catenin, localization in adherens junctions is critical for intercellular adhesion and communication, and nuclear localization is responsible for activating downstream target genes for cell cycle/proliferation [54]. The expression of β-catenin was studied using HCC patients and revealed that decreased expression in the membrane and increased expression in the cytoplasm and nucleus correlated to the development of cirrhosis and HCC [68]. Our data indicate that AURKA(WT) promotes accumulation of cytoplasmic β-catenin, while AURKA(V352I) increases the nuclear β-catenin and decreases membrane form β-catenin. These data imply that tumorigenesis in AURKA(V352I) is different from AURKA(WT).

Sorafenib has been approved by the FDA for advanced HCC. However, it has no effect on AURKA(WT) and AURKA(V352I) transgenic fish. Nonetheless, Sorafenib effectively decreased HCC formation induced by HBx and Src overexpression in p53 mutant transgenic fish [58]. It was revealed that hesperidin supplementation initiated apoptosis via targeted inhibition of constitutively activated Aurora-A-mediated PI3K/Akt/GSK-3β and mTOR pathways in colorectal cancer [69]. This therapeutic approach will be tested in AURKA transgenic fish. Although the mutation of AURKA(V352I) is a rare event, β-catenin activation induced HCC frequently turned malignant. Significantly, this study shows that personalized medicine should be based on genomic makeup, and the variations of the AURKA and downstream signaling pathways should be considered in order to find effective, personalized therapeutics. Using the AURKA(WT) and AURKA(V352I) transgenic zebrafish model, we could test the anti-cancer effect of multiple kinase inhibitors and combinational therapeutics.

## 4. Materials and Methods

### 4.1. Generate Tol2 Expression Constructs

Three transgenic zebrafish were used in this study, Tg(fabp10a: AURKA(V352I), myl7: EGFP)-abbreviated as AURKA(V352I), Tg(fabp10a: AURKA(WT), myl7: EGFP), abbreviated as AURKA(WT), and Tg(fabp10a:EGFP-mCherry), abbreviated as mC (as a control zebrafish). The expression constructs were generated by Tol2 gateway transgenesis using the MultiSite Gateway^®^ Three-Fragment Vector Construction Kit [70]. Liver-specific fatty acid binding protein 10a (f*abp10a*) drives the expression of AURKA and the EGFP-mCherry fusion gene in the control, and the heart-specific myosin light chain 7 (*myl7*) drove the expression of EGFP in the heart as a transgene marker. We first designed the forward primer: attB1-AURKA-F (sequence below: GGGGACAAGTTTGTACAAAAAAGCAGGCTATGGACCGATCTAAAGAAAACTG, the underlined nucleotide ATG is the translation initiation site), and the reverse primer: attB2-AURKA-R (sequence below: GGGGACCACTTTGTACAAGAAAGCTGGGTCTAAGACTGTTTGCTAGCTGATTCTT, the underlined nucleotide CTA is the reverse sequence of the termination codon TAG). The attB1-AURKA-F and the attB2-AURKA-R were used for PCR to amplify the AURKA cDNA sequence. The 1270 bp PCR product was used for the BP reaction with the pDONR221 vector to generate pME-AURKA(WT) and pME-AURKA(V352I) entry clones. Then, the final expression construct was generated by LR reaction using three entry clones, including p5E-fabp10a (contains *att*L4 and *att*R1 surrounding liver specific promoter fabp10a), pME-AURKA: pME-AURKA(WT) or AURKA(V352I) (contains *att*L1 and *att*L2 surrounding AURKA, and p3E-pA (contains *att*R2 and *att*L3 surrounding polyA sequence), together with the destination vector, pDESTCG5. The HE expression constructs were confirmed by sequencing to verify the correct linkage of the fabp10a promoter with AURKA cDNA and polyA with a heart specific florescent in the vector, and the expressions the constructs were microinjected into one-cell stage embryos, developed from the AB strain of *Danio rerio*, with transposase RNA. The AB strain was obtained from the Zebrafish International Resource Center (ZIRC, Eugene, OR, USA).

### 4.2. Zebrafish Maintenance

Zebrafish were maintained in the Zebrafish Core Facility at National Health Research Institute (NHRI) with a controlled light cycle of 14 h of light/10 h of dark at 28 °C and were fed dry food or Artemia three times every day. All zebrafish experiments were approved by the Institutional Animal Care and Use Committee (IACUC) of the NHRI and were in accordance with International Association for the Study of Pain guidelines (protocol number: NHRI-IACUC-106120-A). The Taiwan Zebrafish Core Facility at NHRI or TZeNH is a government-funded core facility, and since 2015, it has been accredited by the Association for Assessment and Accreditation of Laboratory Animal Care International (AAALAC) [71].

### 4.3. Liver Tissue Collection and Paraffin Section

Ten fish from each transgenic line were sacrificed at 3, 5, 7, 9, and 11 months, and liver specimens were collected and divided into two parts. One part was frozen immediately for RNA extraction and Reverse Transcription-Polymerase Chain Reaction (RT-PCR), while the other part was paraffin fixed and sectioned for sequential immunohistochemistry (IHC) analysis and staining. For paraffinized sections, the liver tissues were immersed in 10% formaldehyde (Sigma-Aldrich Inc., St. Louis, MO, USA) at 4 °C for 24 h and transferred to 75% ethanol at 4 °C for long-term storage. The tissues were embedded and sectioned at 5 μm thickness mounted coated poly-L-lysine slides in paraffin. The sections were stained with HE and IHC staining. The embedding, section, and HE stain were performed by the Pathology core facility in National Health Research Institutes.

### 4.4. Hematoxylin and Eosin Staining

The slides were brought to room temperature and the sections were covered with a liquid blocker (PapPen) to prevent incubation solution evaporation, and incubated with 4% paraformaldehyde dissolved in PBS (pH 7.2) for 5 min and then put in a glass chamber and washed with running tap water for ~1 min to remove the rest of the formaldehyde. After draining the slides on a tissue paper, the slides were incubated with Mayer’s hematoxylin solution for 5 min to stain the nuclei and then put in a glass chamber and washed under warm running tap water for 10 min to change the color of the nuclei to blue. Acetic acid (1:100) was added to the 0.5% Eosin solution (Merck 1.09844.1000), and the slides were incubated for 10 min to stain the fibers red. The slides were washed 3 times for 1 min with double diluted water to remove excess of Eosin in a glass chamber. The slides were drained successively in 70% ethanol for 1 min, 90% ethanol for 30 s, 100% ethanol for 30 s, and Xylene for 30 s and were mounted with 1–2 drops of a xylene-based mounting medium and covered with cover slides and then stored at room temperature.

### 4.5. Immunohistochemistry Staining

Primary antibodies were used for IHC staining, including PCNA (PC10) mouse mAb (CST#2586), phospho-Akt (Ser473) antibody (CST#9271), phospho-mTOR (Ser2448), rabbit mAb (CST#2976), and PTEN rabbit mAb (CST#9559). The parafilm-embedded slides were de-paraffinized by non-xylene and were re-hydrated by ethanol. In addition, the protein of the tissue specimens (that were fixed in paraffin) forms cross-links that mask the antigenic sites in the tissue sections. The masking effect resulted in a weak signal of IHC staining. To prevent the masking effect, the slide was put in an Antigen Retrieval Buffer (10 mM Sodium citrate, pH6.0 + 0.05% polyoxyethylene 20 sorbitan monolaurate (Tween-20, Ref: J T BAKER X251-07)) and incubated for 20 min at 95 °C. Then, the tissue sections were washed in 1X Tris Buffered Saline (TBS: 25 mM Tris, 150 mM NaCl, pH 7.2) for 3 min. However, the failure signals of the IHC staining could have resulted from the high background caused by endogenous peroxidase activity and non-specific binding. Endogenous peroxidase was quenched by incubating in 0.3% H_2_O_2_ (diluted in pure methanol) for 20 min at room temperature, and the slides were washed with 1X TBS for 5 min three times. To reduce non-specific binding of antibodies, the slides were incubated in a blocking buffer (5% goat serum diluted in 1X TBS) for 1 h at room temperature. The sections were covered by the primary antibody diluted in a blocking buffer and were incubated at 4 °C overnight (about 16–18 h). On the next day, the sections covered by the primary antibody were incubated for 1 h at room temperature and washed with TBST (1X TBS + 0.1% Tween-20) for 10 min 3 times. The biotinylated antibodies with secondary antibodies diluted in the blocking buffer were sheltered in the tissue sections for 30 min at room temperature and were washed with TBST for 10 min 3 times. To amplify the signal intensity, these sections were incubated with an avidin–biotin complex (ABC) reagent (Ref: 32054) for 30 min at room temperature and then washed with TBST for 10 min 3 times. DAB (3,3′-diaminobenzidine) detection was performed using an Invitrogen liquid DAB substrate kit. The DAB oxidized in the presence of peroxidase and hydrogen peroxide and produced the deposition of a brown signal. The sections were incubated in DAB reagent and washed with ddH_2_O (distillation-distillation H_2_O) for 3 min 3 times until brown signals appeared. A hematoxylin counterstain was added to the sections for 5 min and then was washed by ddH_2_O for 3 min 3 times. Following dehydration and mounting, they were covered with a coverslip, and examined with polarized light microscopy; pictures were taken with 50×, 100× and 400× magnifying power. Immunohistochemistry scoring was graded based on two parameters: the intensity grade (score: 1–3) and the proportion of positive tumor cells (score: 1–4) [52]. The immunoreactive score (IRS) was obtained by multiplying the intensity grade by the positive proportion score [53].

### 4.6. Total RNA Isolation

Total RNA was isolated using the NucleoSpin^®^ RNA kit (Ref: 740955.50, (MACHEREY-NAGEL INC. Bethlehem, PA, USA). About 30 milligrams of liver samples were added to 350 μL Buffer RA-1 and 3.5 μL β-mercaptoethanol (Sigma-Aldrich Inc., St. Louis, MO, USA), and the mixture was flash-frozen in liquid nitrogen immediately and stored at −80 °C. Frozen samples were thawed slowly before starting with the isolation of RNA. Thawed samples were homogenized by a Bullet Blender^®^ Homogenizer (Standard BBX24, Next Advance, Inc., Troy, NY, USA) with 0.5 mm diameter zirconium oxide beads (ZrOB05, Next Advance, Inc., Troy, NY, USA). To reduce viscosity and clear the lysate, the lysate was filtrated through the NucleoSpin^®^ Filter (violet ring) by centrifuging for one minute at 11,000× *g* (rcf). After discarding the NucleoSpin^®^ Filter (violet ring), 350 μL 70% ethanol was added to the homogenized lysate and mixed by pipetting up and down to adjust RNA binding conditions. The lysate was transferred to the NucleoSpin^®^ RNA Column (light-blue ring) by pipetting several times, and the RNA was bound to the NucleoSpin^®^ RNA Column (light-blue ring) by centrifuging for 30 s at 11,000× *g* (rcf). Then, 350 μL membrane desalting buffer (MDB) was added for desalting the silica membrane by centrifuging for one minute at 11,000× *g* (rcf). For digesting DNA, 95 μL DNase reaction mixture (10% RNase-free DNase and 90% reaction buffer for DNase) was directly applied onto the center of the silica membrane of the column and incubated at room temperature for 15 min. After DNA digestion, three washes were performed by RAW2 and RA3. For inactivating the DNase, 200 μL RAW2 was first added and centrifuged for 30 s at 11,000× *g* (rcf). Sequentially, 600 and 250 μL RA3 were added and centrifuged, respectively, for 30 s and 2 min at 11,000× *g* (rcf). Finally, RNA was eluted in 40 μL RNase-free H_2_O by centrifuging for 1 min at 11,000× *g* (rcf) and stored at −80 °C.

### 4.7. Reverse Transcription-Polymerase Chain Reaction (RT-PCR)

Real-time PCR was performed by the iScriptTM cDNA Synthesis Kit (BIO-RAD, Hercules, CA, USA). According to the protocol, the concentration of RNA samples needed up-to-the-proper condition (100 fg–1 μg per 20 μL) to ensure optimum synthesis efficiency. Before the preparation of RT-PCR, the concentration of RNA samples was normalized to 300 ng per 20 μL. The preparation and condition of the reverse transcription (RT) reaction mixture are as follows. The cDNA samples were stored at −20 °C.

### 4.8. Quantitative Polymerase Chain Reaction (qPCR)

We used KAPA SYBR^®^ FAST qPCR Kits (Kapa Biosystem) for the qPCR experiment. The sequence of primers for qPCR was listed at Table 1. The quantitative polymerase chain reaction contains the following ingredients: 3.8 µL of cDNA (diluted with RNase free water), 1.2 µL of primer mix (2.5 µM of both forward and reverse primer), and 5.0 µL of 2× SybrGreen Mix, all of which were added to the 384-well qPCR plate. Template DNA was diluted to 100× with RNase-free H_2_O and was loaded into the 384-well qPCR plate. Forward and reverse primers were diluted to 100× with ddH_2_O and mixed with 2× KAPA SYBR^®^ FAST Master Mix (2×) ROX Low. Because SybrGreen is photosensitive, the mixture had to be added last. Once the qPCR reagent was set up, an optical adhesive cover was used to seal the plate with a sealing comb.

The thermal cycling program of quantitative polymerase chain reaction in the ABI HT-7900 machine was set as follows: stage I: 50 °C—2 min, 95 °C—5 min, 4 °C—forever; stage II: 95 °C—10 min; stage III: 95 °C—15 s, 60 °C—1 min (40 cycles); stage IV: 95 °C—15 s, 60 °C—15 s, 95 °C—15 s. Triplicated analysis was performed for each qPCR sample to reduce technical errors. The Ct was normalized to the internal control (β-actin) to obtain the ΔCt. The gene expression ratio between the experimental and control groups was calculated by the comparative Ct (ΔΔCt) method [72]. The relative expression ratio (fold) was calculated based on ΔΔCt = ΔCt (experimental) − ΔCt (control), and fold change = 2^−ΔΔCt^. The mean value and the standard deviation (SD) of the triplicated experiment were calculated.

### 4.9. Statistical Analysis

Statistical analysis of the results was performed using Prism GraphPad and two-tailed Student’s *t*-test. In all statistical analyses, a *p*-value <0.05 was considered to be statistically significant and is shown as: * *p*-value ≤ 0.05; ** *p*-value ≤ 0.01; *** *p* ≤ 0.001; **** *p*-value ≤ 0.0001.

## 5. Conclusions

In conclusion, AURKA(WT) overexpression promotes hepatocarcinogenesis by activating membrane β-catenin, whereas AURKA(V352I) overexpression activates p-AKT, increases nuclear β-catenin, and induces earlier and more severe hepatocarcinogenesis. Development of HCC in AURKA(WT) and AURKA(V352I) is completely different, and precisely understanding the expression level and genotype is necessary for precision personalized medicine. The overexpression of AURKA in mouse MEF cells did not show a transformed phenotype [34] and in transgenic livers led to a low incidence (3.8%) (2 of 52) of hepatic tumor formation after a long latency period in mouse models [35], clearly indicating that AURKA overexpression alone is insufficient to induce carcinogenesis.

## Figures and Tables

**Figure 1 cancers-11-00927-f001:**
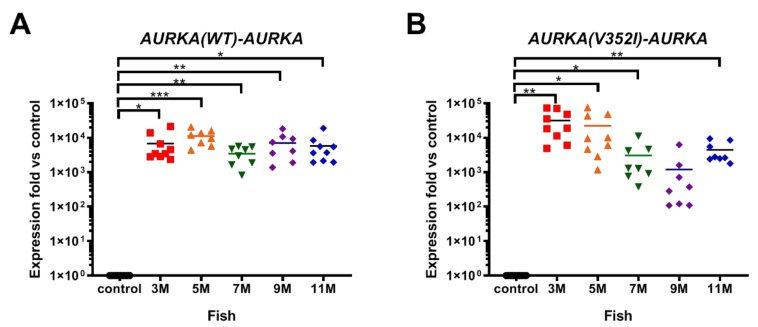
Expression of Aurora A kinase (AURKA) in AURKA(WT) and AURKA(V352I) transgenic zebrafish compared with control fish. qPCR analysis of AURKA in transgenic zebrafish (**A**) Tg(fabp10a:AURKA(WT)-EGFP-mCherry, myl7:EGFP); (**B**) Tg(fabp10a:AURKA(V352I)-EGFP-mCherry, myl7:EGFP). Expression fold compared to control fish (Tg(fabp10a:EGFP-mCherry) is shown in red (3 M), orange (5 M), green (7 M), purple (9 M), and blue (11 M). Statistical analysis of these results was performed using a two-tailed Student’s *t*-test. Asterisks (*) represent the level of significance. * *p*-value ≤ 0.05; ** *p*-value ≤ 0.01; *** *p*-value ≤ 0.001.

**Figure 2 cancers-11-00927-f002:**
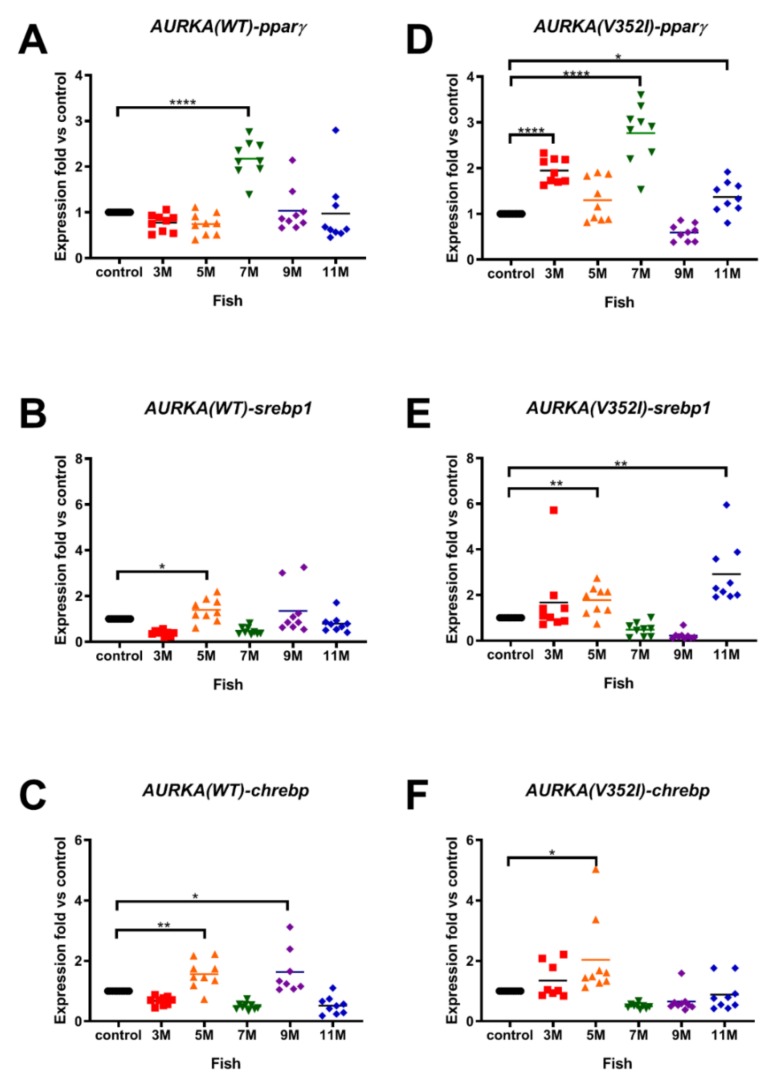
Expression of lipogenic factors (peroxisome proliferator-activated receptor gamma (*pparγ*), sterol regulatory element-binding transcription factor 1 (*srebp1*), carbohydrate-responsive element-binding protein (*chrebp*)) in AURKA(WT) and AURKA(V352I) transgenic fish were higher than control. qPCR analysis of lipogenic factors (**A**,**D**) *pparγ*; (**B**,**E**) *srebp1*; (**C**,**F**) *chrebp* in AURKA(WT) (**A**–**C**) and AURKA(V352I) (**D**–**F**) transgenic zebrafish compared to control fish at different time points. Expression fold compared to control fish Tg(fabp10a:EGFP-mCherry is shown in red (3 M), orange (5 M), green (7 M), purple (9 M), and blue (11 M). Statistical analysis of results was performed using a two-tailed Student’s *t*-test. Asterisks (*) represent the level of significance. * *p*-value ≤ 0.05; ** *p*-value ≤ 0.01; *** *p*-value ≤ 0.001; **** *p*-value ≤ 0.0001.

**Figure 3 cancers-11-00927-f003:**
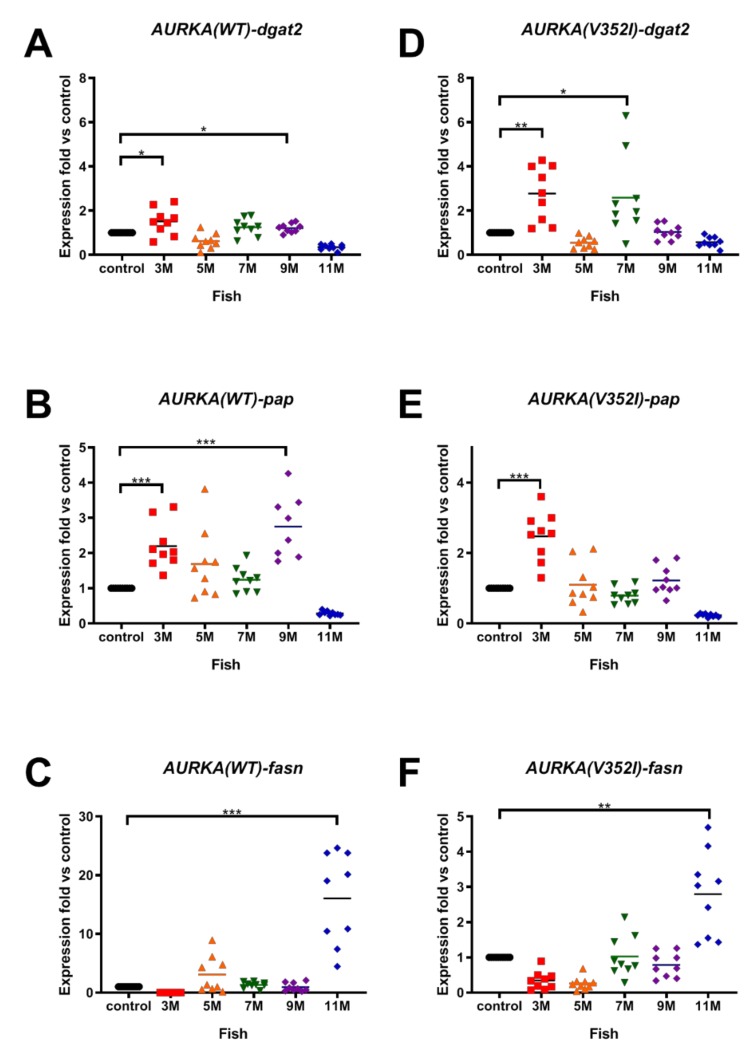
Expression of lipogenic enzymes (diacylglycerol *O*-Acyltransferase 2 (*dgat2*), phosphatidate phosphatase (*pap*), fatty acid synthase (*fasn*)) in AURKA(WT) and AURKA(V352I) transgenic fish were higher than control. qPCR analysis of lipogenic enzyme (**A**,**D**) *dgat2*; (**B**,**E**) *pap*; (**C**,**F**) *fasn* in AURKA(WT) (**A**–**C**) and AURKA(V352I) (**D**–**F**) transgenic zebrafish compared to control fish at different time points. Expression fold compared to control fish (Tg(fabp10a:EGFP-mCherry) is shown in red (3 M), orange (5 M), green (7 M), purple (9 M), and blue (11 M). Statistical analysis of results was performed using a two-tailed Student’s *t*-test. Asterisks (*) represent the level of significance. * *p*-value ≤ 0.05; ** *p*-value ≤ 0.01; *** *p*-value ≤ 0.001.

**Figure 4 cancers-11-00927-f004:**
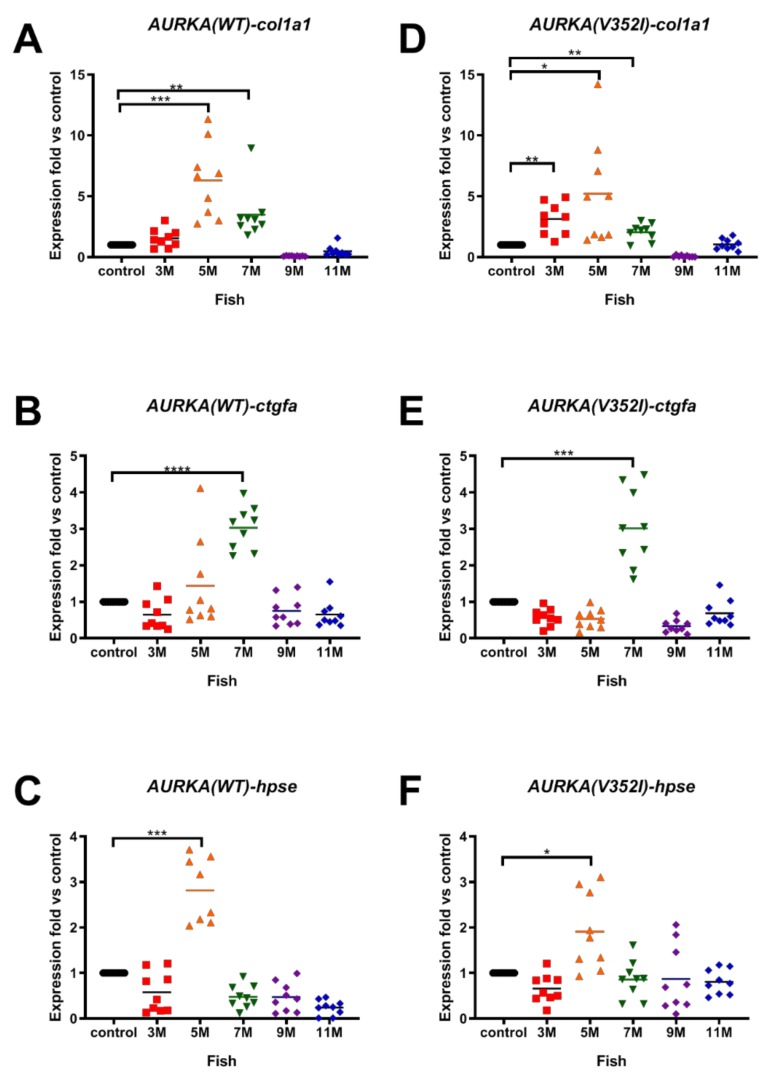
Expression of fibrosis markers (collagen type I alpha 1 (*col1a1*), connective tissue growth factor (*ctgfa*), heparanase (*hpse*)) in AURKA(WT) and AURKA(V352I) transgenic fish compared to control. qPCR analysis of fibrosis markers (**A**,**D**) *col1a1*; (**B**,**E**) *ctgfa*; (**C**,**F**) *hpse* in AURKA(WT) (**A**–**C**) and AURKA(V352I) (**D**–**F**) transgenic zebrafish compared to control fish at different time points. Expression fold compared to control fish (Tg(fabp10a:EGFP-mCherry) is shown in red (3 M), orange (5 M), green (7 M), purple (9 M), and blue (11 M). Statistical analysis of results was performed using a two-tailed Student’s *t*-test. Asterisks (*) represent the level of significance. * *p*-value ≤ 0.05; ** *p*-value ≤ 0.01; *** *p*-value ≤ 0.001; **** *p*-value ≤ 0.0001.

**Figure 5 cancers-11-00927-f005:**
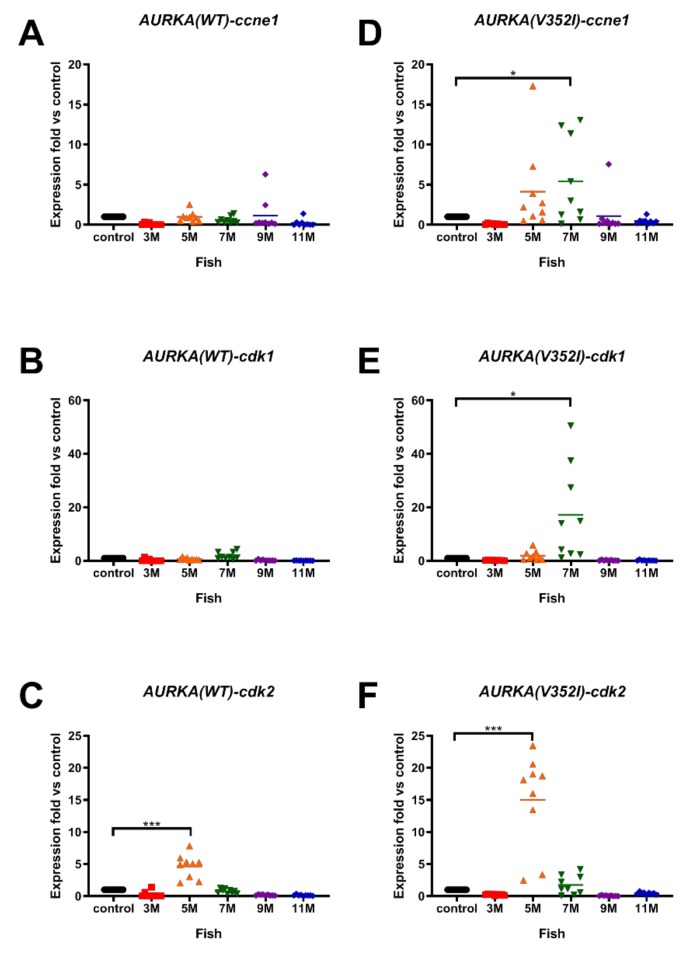
Expression of cell cycle related genes (cycle-related genes/proliferation marker genes G1/S-specific cyclin-E1 (*ccne1*), cyclin-dependent kinase 1 (*cdk1*), cyclin-dependent kinase 2 (*cdk2*)) was higher in AURKA(V352I) than AURKA(WT) transgenic fish. qPCR analysis of cell cycle/proliferation markers (**A**,**D**) *ccne1*; (**B**,**E**) *cdk1*; (**C**,**F**) *cdk2* in AURKA(WT) (**A**–**C**) and AURKA(V352I) (**D**–**F**) transgenic zebrafish compared to control fish at different time points. Expression fold compared to control fish (Tg(fabp10a:EGFP-mCherry) is shown in red (3 M), orange (5 M), green (7 M), purple (9 M), and blue (11 M). Statistical analysis of results was performed using a two-tailed Student’s *t*-test. Asterisks (*) represent the level of significance. * *p*-value ≤ 0.05; *** *p*-value ≤ 0.001.

**Figure 6 cancers-11-00927-f006:**
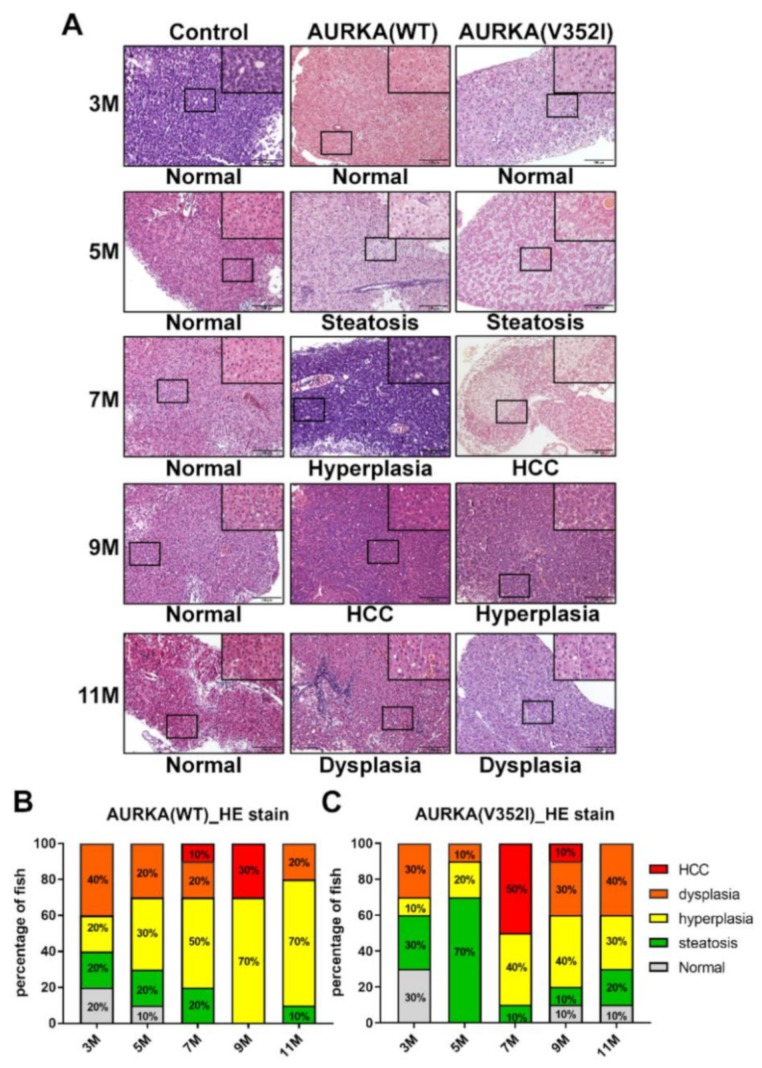
Hematoxylin and Eosin staining reveals that AURKA(V352I) dramatically promotes hepatocellular carcinoma (HCC) at 7 months, and AURKA(WT) promotes HCC at 9 months of age. (**A**) Representative images of HE staining in control, AURKA(WT) and AURKA(V352I) at 3, 5, 7, 9, and 11 months; (**B**,**C**) Statistical analysis of the AURKA(WT) and AURKA(V352I) HE stain results shown as percentage of fish displayed normal (gray), steatosis (green), hyperplasia (yellow), dysplasia (orange), and HCC (red) at different stages.

**Figure 7 cancers-11-00927-f007:**
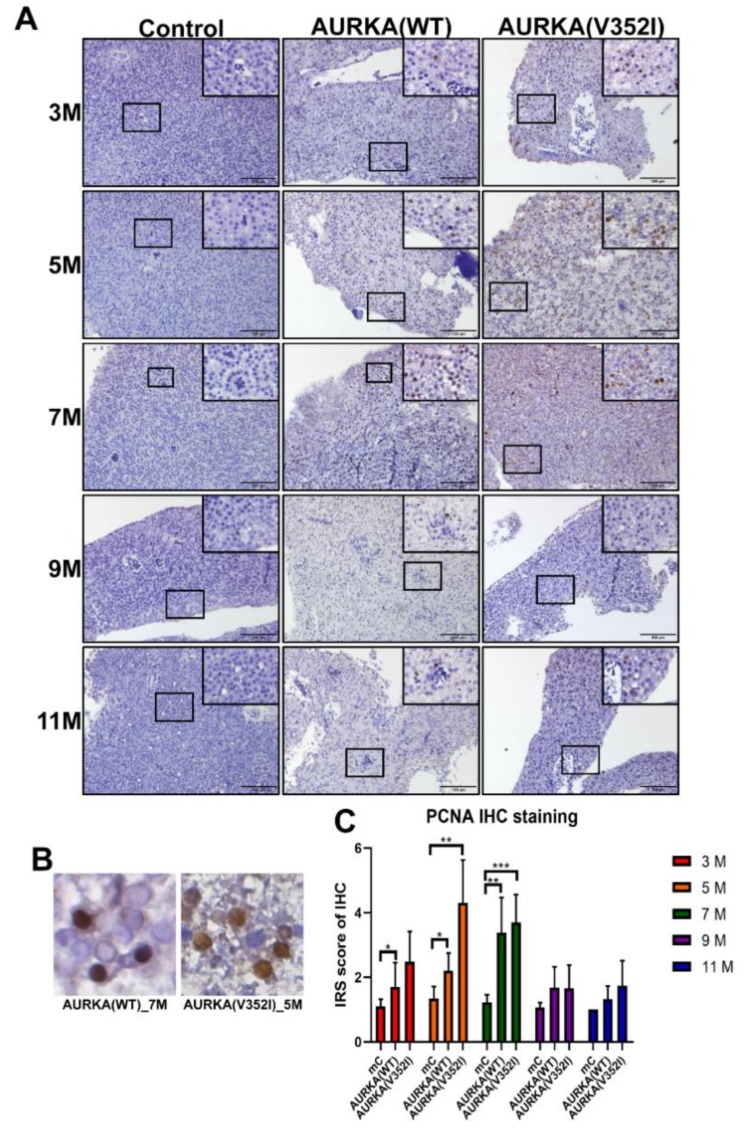
The immunoreactive score (IRS) of immunohistochemistry for the proliferating cell nuclear antigen (PCNA) was greater in AURKA(V352I) and AURKA(WT) than in control fish at 5 and 7 months. (**A**) Representative images of immunohistochemistry (IHC) staining for PCNA in control, AURKA(WT), and AURKA(V352I) at 3, 5, 7, 9, and 11 months; (**B**) Selected enlarged images show the nuclear signals of PCNA from AURKA(WT)—7 M and AURKA(V352I)—5 M; (**C**) Statistical analysis of PCNA immunostaining IRS score at 3, 5, 7, 9, and 11 months. IRS score of control fish (Tg(fabp10a:EGFP-mCherry) abbreviated as mC, AURKA(WT) and AURKA(V352I) is shown in red (3 M), orange (5 M), green (7 M), purple (9 M), and blue (11 M). Statistical analysis of results was performed using a two-tailed Student’s *t*-test. The error bar means standard deviation. Asterisks (*) represent the level of significance. * *p*-value ≤ 0.05; ** *p*-value ≤ 0.01; *** *p*-value ≤ 0.001.

**Figure 8 cancers-11-00927-f008:**
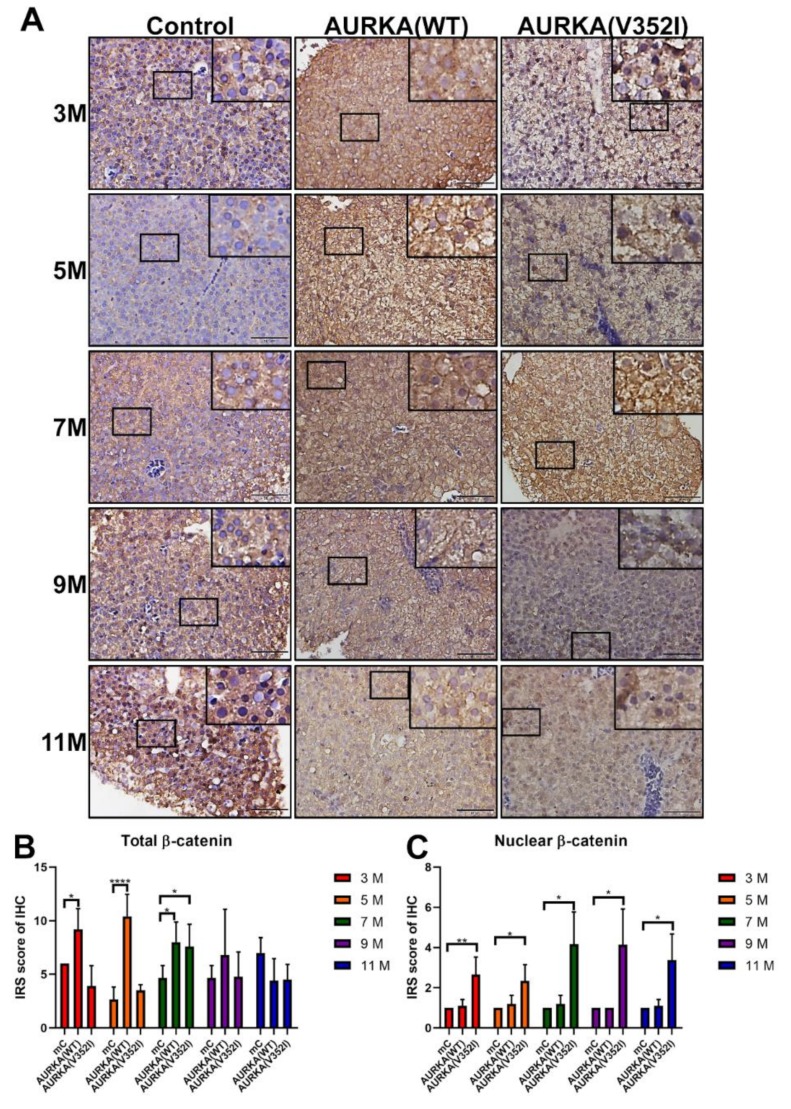
Immunoreactive score of immunohistochemistry for β-catenin reveals that AURKA(V352I) is significantly lower than control and AURKA(WT) at 3, 5, and 7 months. (**A**) Representative images of immunohistochemistry (IHC) staining for β-catenin in control, AURKA(WT) and AURKA(V352I) at 3, 5, 7, 9, and 11 months; (**B**) Statistical analysis of β-catenin immunostaining IRS score at 3, 5, 7, 9, and 11 months; (**C**) Statistical analysis of nuclear β-catenin immunostaining IRS score at 3, 5, 7, 9, and 11 months. IRS score of control fish (Tg(fabp10a:EGFP-mCherry) abbreviated as mC, AURKA(WT) and AURKA(V352I) is shown in red (3 M), orange (5 M), green (7 M), purple (9 M), and blue (11 M). Statistical analysis of results was performed using a two-tailed Student’s *t*-test. The error bar means standard deviation. Asterisks (*) represent the level of significance. * *p*-value ≤ 0.05; ** *p*-value ≤ 0.01; *** *p*-value ≤ 0.001.

**Figure 9 cancers-11-00927-f009:**
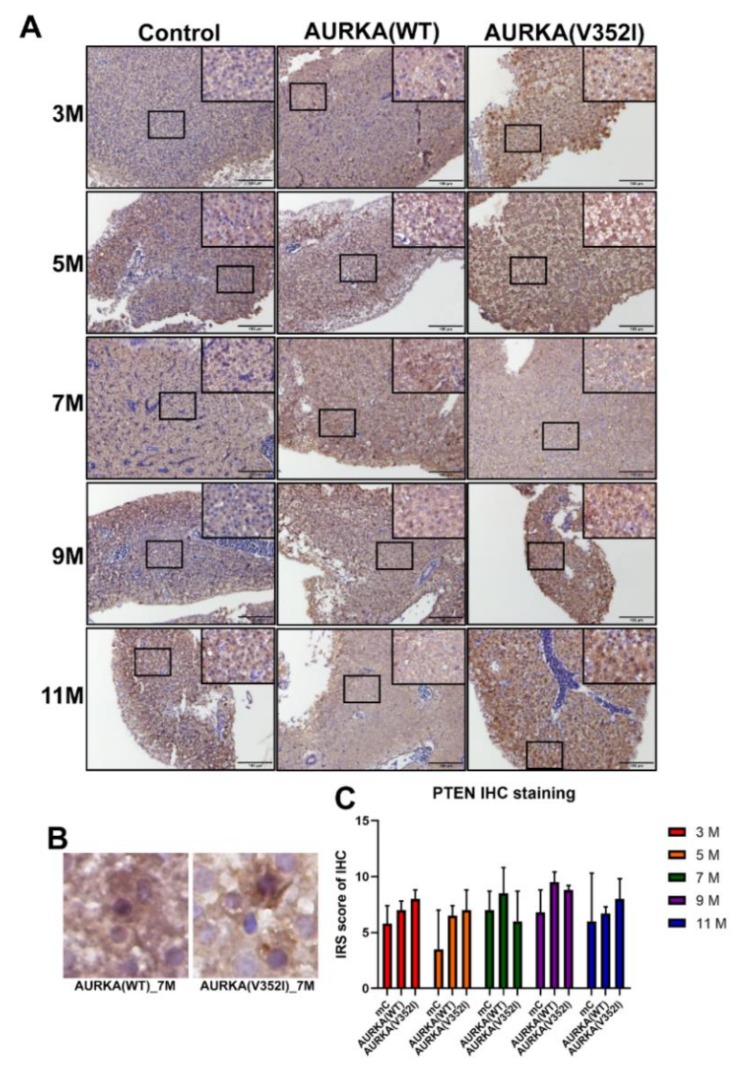
Immunoreactive score of the phosphatase and tensin homologues deleted on chromosome 10 (PTEN). Immunostaining reveals that PTEN shows no difference between control and AURKA transgenic zebrafish. (**A**) Representative images of immunohistochemistry (IHC) staining for PTEN in control, AURKA(WT), and AURKA(V352I) at 3, 5, 7, 9, and 11 months; (**B**) Selected enlarged images show the PTEN signals from AURKA(WT)—7 M and AURKA(V352I)—7 M; (**C**) Statistical analysis of PTEN immunostaining IRS score at 3, 5, 7, 9 and 11 months. IRS score of control fish (Tg(fabp10a:EGFP-mCherry) abbreviated as mC, AURKA(WT) and AURKA(V352I) is shown in red (3 M), orange (5 M), green (7 M), purple (9 M), and blue (11 M). Statistical analysis of results was performed using a two-tailed Student’s *t*-test. The error bar means standard deviation.

**Figure 10 cancers-11-00927-f010:**
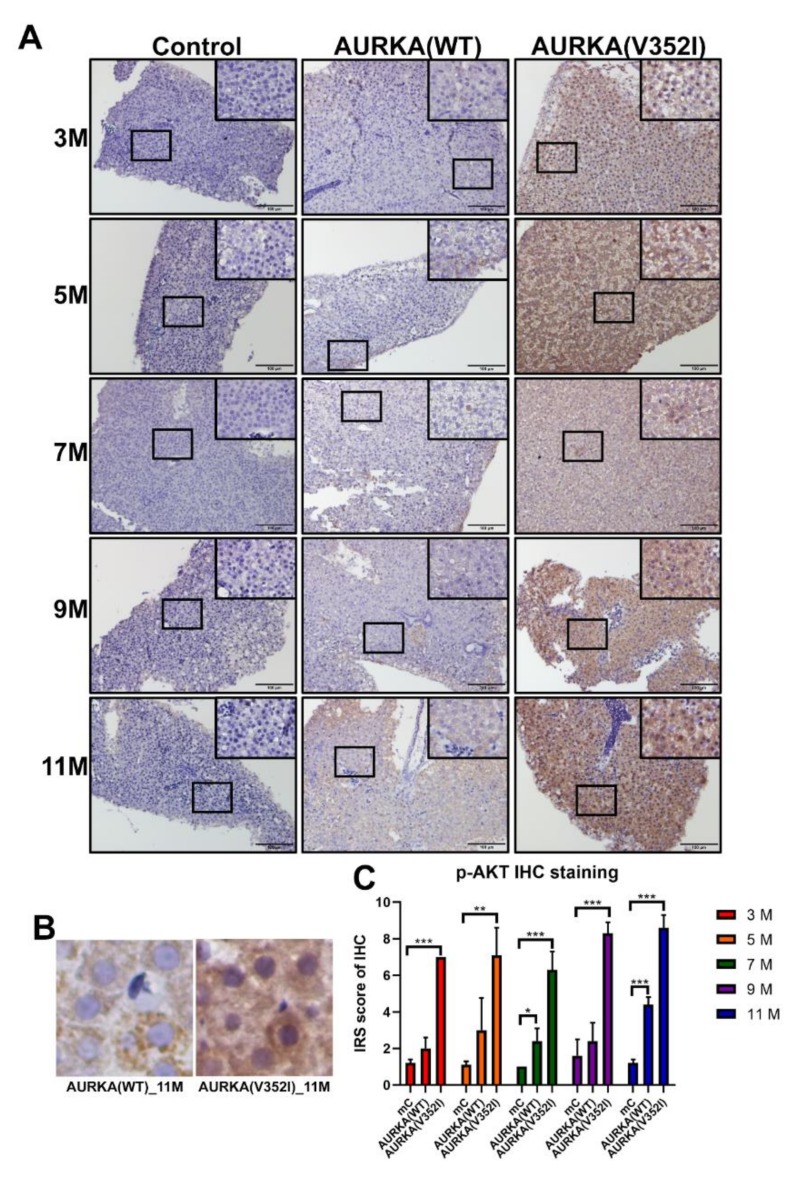
Immunoreactive score of p-Akt immunostaining reveals that Akt is much more significantly activated in AURKA(V352I) than in AURKA(WT). (**A**) Representative images of p-AKT staining for PTEN in control, AURKA(WT), and AURKA(V352I) at 3, 5, 7, 9, and 11 months; (**B**) Selected enlarged images show the p-AKT signals from AURKA(WT)—11 M and AURKA(V352I)—11 M; (**C**) Statistical analysis of p-AKT immunostaining IRS score at 3, 5, 7, 9, and 11 months. The gray, orange, and blue colors represent control, AURKA(WT), and AURKA(V352I), respectively. Statistical analysis of results was performed using a two-tailed Student’s *t*-test. The error bar means standard deviation. Asterisks (*) represent the level of significance. * *p*-value ≤ 0.05; ** *p*-value ≤ 0.01; *** *p*-value ≤ 0.001.

**Figure 11 cancers-11-00927-f011:**
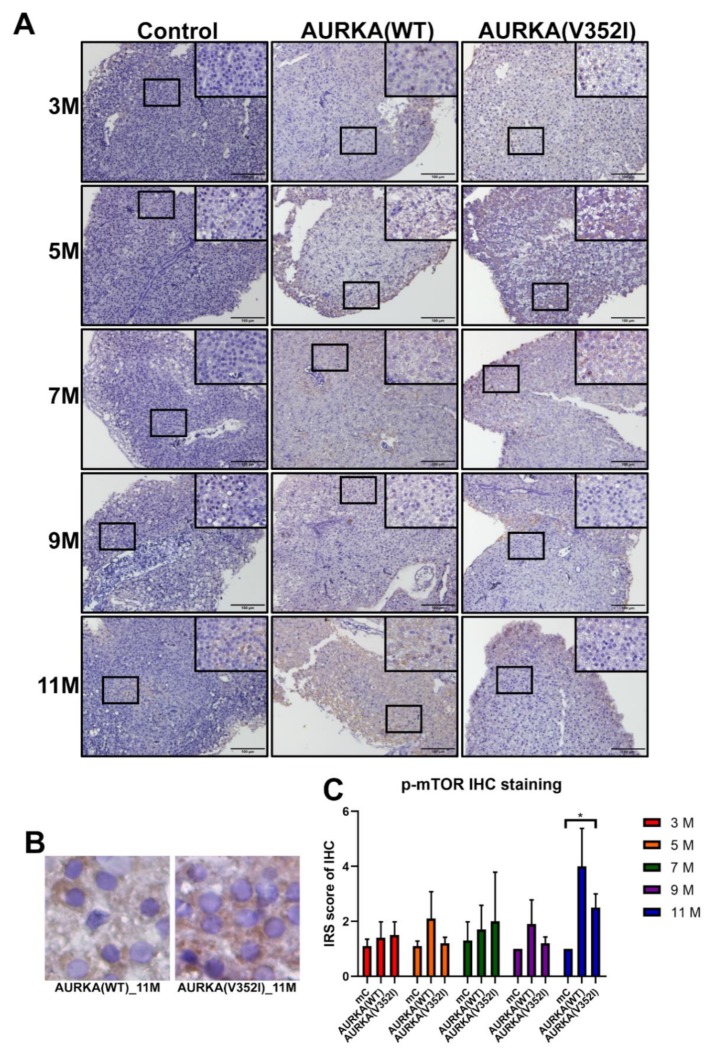
Immunoreactive score of phospho- mammalian target of the rapamycin (mTOR) (at Ser2448, inactive form) immunostaining for the AKT/mTOR pathway reveals that the expression of mTOR has no significant difference between control and AURKA transgenic fish. (**A**) Representative images of p-mTOR staining for PTEN in control, AURKA(WT), and AURKA(V352I) at 3, 5, 7, 9, and 11 months; (**B**) Selected enlarged images show the p-mTOR signals from AURKA(WT)—11 M and AURKA(V352I)—11 M; (**C**) Statistical analysis of p-mTOR immunostaining IRS score at 3, 5, 7, 9, and 11 months. IRS score of control fish (Tg(fabp10a:EGFP-mCherry) abbreviated as mC, AURKA(WT) and AURKA(V352I) is shown in red (3 M), orange (5 M), green (7 M), purple (9 M), and blue (11 M). Statistical analysis of results was performed using a two-tailed Student’s *t*-test. The error bar means standard deviation. Asterisks (*) represent the level of significance. * *p*-value ≤ 0.05.

**Table 1 cancers-11-00927-t001:** Primer sequence of qPCR for human *AURKA*, zebrafish *actin*, lipogenic factors (*pparγ*, *srebp1*, *chrebp*), lipogenic enzymes (*dgat2*, *pap*, *fasn*), fibrosis markers (*col1a1*, *ctgfa*, *hpse*) and cell cycle related genes (*ccne1*, *cdk1*, *cdk2*).

Gene Name	Primer Name	Sequence (5′ to 3′)
*AURKA*	Q-*AURKA*-F	TGGAATATGCACCACTTGGA
	Q-*AURKA*-R	ACTGACCACCCAAAATCTGC
*actin*	Q-*actin*-F	CTCCATCATGAAGTGCGACGT
	Q-*actin*-R	CAGACGGAGTATTTGCGCTCA
*pparγ*	*Q-pparγ*-	GGTTTCATTACGGCGTTCAC
	*Q-pparγ*-R	TGGTTCACGTCACTGGAGAA
*srebp1*	Q-*srebp1*-F	CATCCACATGGCTCTGAGTG
	Q-*srebp1*-R	CTCATCCACAAAGAAGCGGT
*chrebp*	Q-*chrebp*-F-2	GGAGATGGACTCGCTCTTTG
	Q-*chrebp*-R-2	GCAGAGGCTCAAAAGTGTCC
*dgat2*	Q-*dgat2*-F	TGGGGCTTTTTGTAACTTCG
	Q-*dgat2*-R	TCTTCCTGGTGCACAGTCC
*pap*	Q-*pap*-F	CAGTTCTTCCTGATTGCTGC
	Q-*pap*-R	TCCTCAAAGCTTAGTTCGGG
*fasn*	Q-*fasn*-F	ATCTGTTCCTGTTCGATGGC
	Q-*fasn*-R	AGCATATCTCGGCTGACGTT
*col1a1*	Q-*col1a1*-F	TATTGGTGGTCAGCGTGGTA
	Q-*col1a1*-R	TCCTGGAGTACCCTCACGAC
*ctgfa*	Q-*ctgfa*-F	TGTGTGTTTGGTGGAATGGT
	Q-*ctgfa*-R	GGAGTCACACACCCACTCCT
*hpse*	Q-*hpse*-F	GCTCTGGTTTGGAGCTCATC
	Q-*hpse*-R	GAAATCCCGACCAAGTTGAA
*ccne1*	Q-*ccne1*-F	TCCCGACACAGGTTACACAA
	Q-*ccne1*-R	TTGTCTTTTCCGAGCAGGTT
*cdk1*	Q-*cdk1*-F	CTCTGGGGACCCCTAACAAT
	Q-*cdk1*-R	CGGATGTGTCATTGCTTGTC
*cdk2*	Q-*cdk2*-F	CAGCTCTTCCGGATATTTCG
	Q-*cdk2*-R	CCGAGATCCTCTTGTTTGGA

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
