# Peer review of "A Novel AURKA Mutant-Induced Early-Onset Severe Hepatocarcinogenesis Greater than Wild-Type via Activating Different Pathways in Zebrafish"

_cancers, 2019, doi:10.3390/cancers11070927_

Round 1

Reviewer 1 Report

In the manuscript from Zhong-Liang Su et al., the authors describe the oncogenic function of the AURKA mutant (V352I) on hepatocellular carcinoma (HCC). They show convincing data for that hepatocarcinogenesis was developed earlier in mutant AURKA zebrafish and was more aggressive  than observed in wild-type AURKA. They have also investigated the effects of mutant AURKA on cell proliferation, lipgenesis, fibrotic markers and some signaling pathways including beta-catenin, AKT and mTOR. They investigated a possible effect of treatment with Sorafenib, a multikinase inhibitor, which is the only FDA-approved drug to treat advanced stage HCC patients. They did not observe any effect of treatment by the use of Sorafenib. Taken together I find this manuscript to present interesting and important data.  

Author Response

We thank the reviewer for his/her comments, we appreciate your support.

Reviewer 2 Report

The authors have carried out a systematic and detailed study regarding the role of AURKA overexpression in HCC development and especially focused on differences between AURKA wt and V352I mutant using zebrafish model. However, there are many observations not explained/discussed well in the text.

Major points:

 1. The expression of lipogenic enzyme and lipogenic factor were much earlier and higher in AURKA(V352I) than in AURKA(WT) transgenic fish: Fig 2. the expression of PPARg, SREBP1 and CHREBP is shown in the liver of wt and mutant fish at various time points however the fluctuations are not clear and makes the argument less convincing. PPARg is high in 7M in wt but it goes down later (fig 2A), likewise SREBP1 is high only in 5M (fig 2B) and CREBP levels are quite fluctuating with time bring high in 5M and 9M only (fig 2C). Is it expected that these genes get downregulated as the disease progress ? there is no explanation for these expression patterns ? Likewise for the mutant cases, PPARg is lowest in 9M (fig2D), SREBP1 is fluctuating (2E) and CHREBP is up only till 5M (2F). it will be nice to add an explanation of these patterns in the discussion.

2. Similar to Fig 2, expression patterns of Fig 3 and 4 genes should also be explained. Especially, it is important to discuss why there is a decrease in expression at a later timepoint.

3. Fig 8, b-catenin expression in control, wt and mutant zebrafish liver is shown b-catenin was found to be lowest in the mutant cases. however, how IRS score is determined should be defined explicitly in the text. also, it has been argued that this decrease could be just die to decrease in membrane-bound b-catenin and actual transcriptional activity of b-catenin is high. In this case, authors should specifically score nuclear b-catenin staining as well. 

4. Multiple kinase inhibitors cannot prevent cancer formation in AURKA (WT) and AURKA (V352I) transgenic fish: in this experiment it is unclear how the effective dose of Sorafenib is determined. There should have been a parallel line of HCC due to other reason than AURKA which is known to respond to Sorafenib so as  to prove the effectiveness of the treatment.

5. The authors have focused on the role of AURKA V352I mutation in HCC, however this is a very rare mutation. Hence, overall significance of the study should be made clear.

Minor point:

IHC images: higher magnification and better resolution images should be provided.

in the methods section, criteria for IRS score for each staining should be explained.

Author Response

Reviewer #2: English language and style are fine/minor spell check required.

Comments: The authors have carried out a systematic and detailed study regarding the role of AURKA overexpression in HCC development and especially focused on differences between AURKA wt and V352I mutant using zebrafish model. However, there are many observations not explained/discussed well in the text.

Major points:

1. The expression of lipogenic enzyme and lipogenic factor were much earlier and higher in AURKA(V352I) than in AURKA(WT) transgenic fish: Fig 2. The expression of PPARg, SREBP1 and CHREBP is shown in the liver of wt and mutant fish at various time points however the fluctuations are not clear and makes the argument less convincing. PPARg is high in 7M in wt but it goes down later (fig 2A), likewise SREBP1 is high only in 5M (fig 2B) and CREBP levels are quite fluctuating with time bring high in 5M and 9M only (fig 2C). Is it expected that these genes get downregulated as the disease progress? There is no explanation for these expression patterns? Likewise for the mutant cases, PPARg is lowest in 9M (fig2D), SREBP1 is fluctuating (2E) and CHREBP is up only till 5M (2F). It will be nice to add an explanation of these patterns in the discussion.

Thanks to reviewer for the constructive comments. Yes, from our independent transgenic lines, we do observe the expression of lipogenic factors were downregulated at later time points. Those expression pattern were due to the progression of hepatocarcinogenesis, those fish developed steatosis at earlier time points accompanied with upregulation of lipogenic factors, and advanced into HCC at later time points with the upregulation of cell cycle/proliferation markers and downregulation of lipogenic factors/enzyme. We have added an explanation of these patterns in the discussion section.

2. Similar to Fig 2, expression patterns of Fig 3 and 4 genes should also be explained. Especially, it is important to discuss why there is a decrease in expression at a later time point.

Thanks to reviewer for the comments. The expression patterns of Fig 3 are lipogenic enzymes, and Fig 4 are fibrosis related genes. Those expression patterns were downregulated at later time points due to the progression of hepatocarcinogenesis. We have added an explanation of these patterns in the discussion section.

3. Fig 8, b-catenin expression in control, wt and mutant zebrafish liver is shown b-catenin was found to be lowest in the mutant cases. However, how IRS score is determined should be defined explicitly in the text. Also, it has been argued that this decrease could be just die to decrease in membrane-bound b-catenin and actual transcriptional activity of b-catenin is high. In this case, authors should specifically score nuclear b-catenin staining as well. 

Thanks to reviewer for the constructive comments. We have mentioned how the IRS score was determined in the section 2.6, page 10 line 259-262. We explained briefly at section 2.7. We have re-scored the nuclear b-catenin staining for total b-catenin and nuclear b-catenin.

4. Multiple kinase inhibitors cannot prevent cancer formation in AURKA (WT) and AURKA (V352I) transgenic fish: in this experiment it is unclear how the effective dose of Sorafenib is determined. There should have been a parallel line of HCC due to other reason than AURKA which is known to respond to Sorafenib so as to prove the effectiveness of the treatment.

Thanks to reviewer for the constructive comments. The effective dose of Sorafenib was determined based on human dosage, and it has been proved to be effective in other transgenic fish lines as published previously. Sorafenib and two other multiple kinase inhibitors (BPR1J419S1 and BPR1J420S1) have been proved to prevent the HCC in HBx,src,p53-/+ transgenic fish [Cancers, 2019, Identification of Novel Anti-Liver Cancer Small Molecules with Better Therapeutic Index than Sorafenib via Zebrafish Drug Screening Platform]. We have added an explanation of these patterns in the section 2.9.

5. The authors have focused on the role of AURKA V352I mutation in HCC, however this is a very rare mutation. Hence, overall significance of the study should be made clear.

Thanks to reviewer for the comments, although the mutation is rare, it did happen in some cancer patients. The significance of this study is personalized medicine should be based on the genomic makeup, the variations of the AURKA should be considered, in order to find a better therapeutics. We have added an explanation of these patterns in the discussion section.

Minor point:

IHC images: higher magnification and better resolution images should be provided.

In the methods section, criteria for IRS score for each staining should be explained.

Thanks to reviewer for the comments, we have replaced with higher resolution images for IHC, and specified the criteria for IRS score in the results and materials and methods section.

In this revised version, we carefully checked and corrected all the Grammar and conceptual errors. The English was corrected by MDPI English editing service.

Reviewer 3 Report

This study uses zebrafish to explore the impact of a human mutation in AURKA on hepatocarcinogenesis by generating zebrafish stably overexpressing WT and mutant AURKA specifically in the liver. Overexpression of b-catenin was seen in WT but not mutant and overexpression of AKT was seen in mutant but not WT, suggesting potential differences in hepatocarcinogenesis, although these were not explored. It is also proposed as a model for drug testing.

Comments:

Figure 1

·      Expression of transgenes is reported as “similar” between WT and mutant yet this graph shows a 3-4 fold difference in AURKA expression at 3M and 7-fold difference at 9M. Could this account for some of the differences between WT and mutant in the studies presented?

Biologically independent repeats with the same transgenic could help here, TG2 has even higher expression levels of AURKA compared to WT, so it is not appropriate when trying to compare mutant and WT phenotypes.

Figure 2

There appears to be a difference between WT and mutant only for ppar-g. Independently confirmed by TG2 expts.

Figure 3

There don’t appear to be any notable differences between WT and mutant for lipogenic enzymes pap and dgat2 (~1.8x vs 2.4x for dgat2 and ~2.2x for both WT and mutant for pap). Confirmed by TG2 expts. Fasn is 18-fold higher than control in WT compared to 3-fold higher in mutant. This is not tested with TG2. On what is the claim that the mutant induces lipogenic enzymes sooner and higher based?

Figure 6

It is curious that some fish regressed at 9-11M in mutant group.

Fig 6B is incorrectly labeled at 5M: the numbers do not add up to 100%

Figure 11

The claim is that mTOR is significantly higher in mutant compared to WT whereas it appears to be lower.

Further detail regarding how the experiments described in section 2.9 were done would be helpful.

General comment

EGFP-mCherry fusion: what does this mean?

Author Response

Reviewer #3: Extensive editing of English language and style required.

Thanks to reviewer for the comments, we already had English Editing Service edited the entire manuscript. We also check the English and spelling errors by ourselves.

Comments:

This study uses zebrafish to explore the impact of a human mutation in AURKA on hepatocarcinogenesis by generating zebrafish stably overexpressing WT and mutant AURKA specifically in the liver. Overexpression of b-catenin was seen in WT but not mutant and overexpression of AKT was seen in mutant but not WT, suggesting potential differences in hepatocarcinogenesis, although these were not explored. It is also proposed as a model for drug testing.

Figure 1

Expression of transgenes is reported as “similar” between WT and mutant yet this graph shows a 3-4 fold difference in AURKA expression at 3M and 7-fold difference at 9M. Could this account for some of the differences between WT and mutant in the studies presented?

Thanks to reviewer for the constructive comments. The expression of lipogenic factors/enzymes and fibrosis markers were downregulated at later time points. Those expression pattern were due to the progression of hepatocarcinogenesis, those fish developed steatosis at earlier time points accompanied with upregulation of lipogenic factors, and advanced into HCC at later time points with upregulation of cell cycle/proliferation markers and downregulation of lipogenic factors/enzymes and fibrosis markers. We have added an explanation of these patterns in the discussion section.

Biologically independent repeats with the same transgenic could help here, TG2 has even higher expression levels of AURKA compared to WT, so it is not appropriate when trying to compare mutant and WT phenotypes.

Thanks to reviewer for the comments. TG1 and TG2 are two independent lines of AURKA(V352I), we also have two other independent lines and observe the similar patterns.

Figure 2

There appears to be a difference between WT and mutant only for ppar-g. Independently confirmed by TG2 expts.

Thanks to reviewer for the comments. Although we observed pparg expression is higher in mutant, other lipogenic factors did not appear to be any notable differences comparing the AURKA(WT) and AURKA(V352I), indicating the AURKA(WT) and AURKA(V352I) can induce steatosis. We have re-written the statement.

Figure 3

There don’t appear to be any notable differences between WT and mutant for lipogenic enzymes pap and dgat2 (~1.8x vs 2.4x for dgat2 and ~2.2x for both WT and mutant for pap). Confirmed by TG2 expts. Fasn is 18-fold higher than control in WT compared to 3-fold higher in mutant. This is not tested with TG2. On what is the claim that the mutant induces lipogenic enzymes sooner and higher based?

Thanks to reviewer for the advice. The expression of lipogenic enzyme does not appear to be much different. We have re-written the statement.

Figure 6

It is curious that some fish regressed at 9-11M in mutant group.

Fig 6B is incorrectly labeled at 5M: the numbers do not add up to 100%

The tumor in transgenic zebrafish will regress due to self-healing of zebrafish. We have added an explanation in the section 2.5. Thanks to reviewer for the careful checking, we have corrected the label in the Fig6B as 5M.

Figure 11

The claim is that mTOR is significantly higher in mutant compared to WT whereas it appears to be lower.

Thanks to reviewer for careful checking, we have re-written the statement. There was no obvious difference for the p-mTOR between AURKA (V352I) and AURKA (WT), except that at 11 months it was significantly higher in AURKA (WT) (Figure 11).

Further detail regarding how the experiments described in section 2.9 were done would be helpful.

Thanks to reviewer for the constructive comments. We have added more detail experimental description on section 2.9. The effective dose of Sorafenib was determined based on human dosage, and it has been proved to be effective in other transgenic fish lines as published previously.

General comment

EGFP-mCherry fusion: what does this mean?

Thanks to reviewer for the comments. Liver-specific fatty acid binding protein 10a (fabp10a) drives the expression of EGFP-mCherry fusion gene in control, the EGFP-mCherry fusion means the gene was a fusion between EGFP and mCherry, under the control of liver specific promoter (fabp10a).

In this revised version, we carefully checked and corrected all the Grammar and conceptual errors. The English was corrected by MDPI English editing service.

Round 2

Reviewer 2 Report

the authors are addressed most of the issues and the manuscript can be accepted at its present state for publication.

Author Response

We thank the reviewer for his/her comments, we appreciate your support. We had check the spell and had English Editing Service edited the entire manuscript.

Reviewer 3 Report

The authors have addressed several of the points. Below are some remaining points that should be addressed:

Figure 1

Expression of transgenes is reported as “similar” between WT and mutant yet this graph shows a 3-4 fold difference in AURKA expression at 3M and 7-fold difference at 9M. 

Respectfully, this still has not been addressed. Expression is only "similar" at 5M, 7M and 11M. At 3M and 9M there are marked differences in expression:  3-4 fold higher AURKA expression in mutant at 3M and 7-fold lower AURKA expression in mutant at 9M compared to WT

        Figure 2

This has been addressed.

        Figure 3

This has been partially addressed. 

Lines 188-198 need to be changed to reflect the new conclusion.

       Figure 6 and Figure 11

This has been addressed.

p.p1 {margin: 0.0px 0.0px 0.0px 0.0px; font: 7.5px Times} span.s1 {color: #b5082d} span.s2 {font: 9.0px 'Times New Roman'}

"we fed the 5-month-old transgenic fish by gavage feeding with these drugs for one month"

Please insert "as published [ref]" or state the dosage.

·      Significance is incorrectly defined in all places.

*: 0.01<p-value0.05; **: 0.001< p-value0.01;

Should read: *:p-value0.05; **:p-value0.01;

Author Response

The authors have addressed several of the points. Below are some remaining points that should be addressed:

Figure 1

Expression of transgenes is reported as “similar” between WT and mutant yet this graph shows a 3-4 fold difference in AURKA expression at 3M and 7-fold difference at 9M.

Respectfully, this still has not been addressed. Expression is only "similar" at 5M, 7M and 11M. At 3M and 9M there are marked differences in expression:  3-4 fold higher AURKA expression in mutant at 3M and 7-fold lower AURKA expression in mutant at 9M compared to WT

Thanks to reviewer for the comments. We have modified the result as “ Although the AURKA mRNA expression level in AURKA (WT) transgenic fish (Figure 1A) was 3-4 fold lower than mutant AURKA transgenic fish (Figure 1B) at 3M, the levels are similar at 5M, 7M and 11M, and interestingly the level was 7-fold lower in AURKA(V352I) compared to AURKA(WT) at 9M.”

Figure 2

This has been addressed.

Lines 188-198 need to be changed to reflect the new conclusion.

Thanks to reviewer for the suggestions. We have modified the result as “The TG2 has higher AURKA expression (Figure S2) and exhibited much higher pparγ expression level compared to TG1 (Figure S3). Although we observed pparγ expression is higher in mutant, other lipogenic factors (srebp1 and chrebp) did not appear to be any notable differences comparing the AURKA(WT) and AURKA(V352I), also confirmed by TG2.”

         Figure 3

         This has been partially addressed.

Thanks to reviewer for the suggestions. We have modified the result as “There don’t appear to be any notable differences between WT and V352I mutant for the expression of lipogenic enzymes pap and dgat2, independently confirmed by another V352I mutant (Figure S4). In V352I mutant, the expression of lipogenic enzyme and lipogenic factor increased at 3 months. In WT, the expression of lipogenic enzyme and lipogenic factor increased from 5 to roughly 7 months. In summary, our results suggest that overexpression both AURKA(WT) and AURKA(V352I) induced expression of lipogenic factors and enzymes, indicating the AURKA(WT) and AURKA(V352I) can induce steatosis.”

         Figure 6 and Figure 11

This has been addressed.

Thanks.

"we fed the 5-month-old transgenic fish by gavage feeding with these drugs for one month"

Please insert "as published [ref]" or state the dosage.

Thanks to reviewer for the comments. We have modified the result as “We fed the 5-month-old transgenic fish by gavage feeding with these drugs for one month as published [58]”

        Significance is incorrectly defined in all places.

*: 0.01<p-value0.05; **: 0.001< p-value0.01;

Should read: *:p-value0.05; **:p-value0.01;

Thanks to reviewer for the comments.

We have modified legend of all figures as: *:p-value0.05; **:p-value0.01;

Round 3

Reviewer 3 Report

All queries have now been addressed.